# CAPE: Channel-Attention-Based PDE Parameter Embeddings for SciML

## ABSTRACT

Scientific Machine Learning (SciML) is concerned with the development of machine learning methods for emulating physical systems governed by partial differential equations (PDE). ML-based surrogate models substitute inefficient and often non-differentiable numerical simulation algorithms and find multiple applications such as weather forecasting, molecular dynamics, and medical applications. While a number of ML-based methods for approximating the solutions of PDEs have been proposed in recent years, they typically do not consider the parameters of the PDEs, making it difficult for the ML surrogate models to generalize to PDE parameters not seen during training. We propose a new channel-attention-based parameter embedding (CAPE) component for scientific machine learning models and a simple and effective curriculum learning strategy. The CAPE module can be combined with any neural PDE solvers allowing them to adapt to unseen PDE parameters without harming the original models' ability to find approximate solutions. The curriculum learning strategy provides a seamless transition between teacher-forcing and fully auto-regressive training. We compare CAPE in conjunction with the curriculum learning strategy using a PDE benchmark and obtain consistent and significant improvements over the base models. The experiments also show several advantages of CAPE, such as its increased ability to generalize to unseen PDE parameters without substantially increasing inference time and parameter count. An implementation of the method and experiments are available at https://anonymous.4open.science/r/CAPE-ML4Sci-145B.

## 1 INTRODUCTION

Many real-world phenomena, ranging from weather forecasts to molecular dynamics and quantum systems, can be modeled with partial differential equations (PDEs). While for some problems the mathematical description of these equations is available, finding its solutions is complex and usually needs some numerical treatments. Numerical simulation methods have been developed for many years and have achieved a high level of accuracy in solving these equations. However, numerical methods are resource intensive and time-consuming even when run on larger supercomputers to obtain sufficiently accurate results. Especially high-resolution and high-dimensional hydrodynamic-type field equations are computationally demanding. The situation becomes even worse if it is necessary to perform simulations with various PDE parameters since a numerical simulation is required for each of the initial conditions and for each PDE parameter's configurations.

Recently, there has been a rapidly growing interest in machine learning methods for the problem of solving PDEs due to their various applications in science and engineering Guo et al. (2016); Lusch et al. (2018); Sirignano & Spiliopoulos (2018); Raissi (2018); Kim et al. (2019); Hsieh et al. (2019); Bar-Sinai et al. (2019); Bhatnagar et al. (2019); Pfaff et al. (2020); Wang et al. (2020); Khoo et al. (2021). For example, several prior studies reported that ML models can estimate solutions more efficiently than classical numerical simulators (Li et al., 2021a; Stachenfeld et al., 2021). Moreover, using neural networks as surrogate models allows us to compute derivatives with respect to the input variables. Differentiable surrogate models allow one to use backpropagation and automatic differentiation to solve the so-called inverse problems which have numerous real-world applications but are difficult to solve using traditional numerical methods (Coros et al., 2013; Allen et al., 2022). A considerable number of papers have shown the advantage of ML-based surrogate models (Li et al., 2020; 2021a; Stachenfeld et al., 2021). The majority of these methods, however, are purely data-

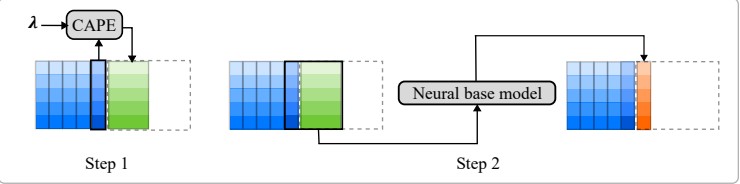

Figure 1: The standard autoregressive approach (left) and the proposed CAPE approach (right) which consists of two interdependent steps.

driven, which does not allow us to change PDE parameters. Although a few models are taking into account PDE parameters, they are tailored to specific neural networks and cannot be used with other state-of-the-art methods. This makes it difficult for the SciML community to develop models with high generalization capability not only for the initial conditions but both for different types of PDEs *and* PDE parameters.

To overcome the shortcomings of existing data-driven SciML models, a straightforward approach would include the PDE parameters as additional input. However, this naive method requires modification of the BASE network which is potentially harmful to its accuracy. An alternative approach attaches an external parameter embedding module to the network. However, there are too many possible module structures and methods to provide the embedded parameter information to the base network, and it is in general non-trivial to select the best one. In this paper, we propose a new and effective parameter embedding module by utilizing the channel-attention method inspired by a numerical solver with the implicit method and style transfer in ML (see Sec. 2.3 and Sec. 2.4). The crucial idea is that a neural network generates intermediate (approximated) field data for future time steps which are then interpolated by a BASE model such as the FNO (Li et al., 2021a) to predict the field data for the next time step. CAPE can be combined with any existing autoregressive neural PDE solvers. Fig. 1 illustrates the proposed CAPE framework.

We make the following contributions. First, we propose a CAPE module which can be combined with any existing neural PDE solvers and can effectively transfer the PDE parameter information to the base network (BASE). Second, we propose a simple but effective curriculum learning strategy that seamlessly bridges the teacher-forcing and auto-regressive methods. Third, we perform extensive experiments using various PDEs with a large number of different parameters evaluating the effectiveness and efficiency of the proposed method in comparison with state-of-the-art methods.

## 2 CAPE: A FRAMEWORK FOR NEURAL PDE SOLVERS

### 2.1 BACKGROUND: PARTIAL DIFFERENTIAL EQUATIONS

Following the notation by Brandstetter et al. (2022), we consider Partial Differential Equations (PDEs) over time dimension $t \in [0, T]$ and over spatial dimensions $\boldsymbol{x} = [x_1, \ldots, x_D] \in \mathbb{X} \subseteq \mathbb{R}^D$ which can be written as

$$\partial_t \boldsymbol{u} = F(t, \boldsymbol{x}, \boldsymbol{u}, \partial_{\boldsymbol{x}} \boldsymbol{u}, \partial_{\boldsymbol{x}, \boldsymbol{x}} \boldsymbol{u}, \ldots), \quad (t, \boldsymbol{x}) \in [0, T] \times \mathbb{X} \tag{1}$$

$$\boldsymbol{u}(0, \boldsymbol{x}) = \boldsymbol{u}^0(\boldsymbol{x}), \ \boldsymbol{x} \in \mathbb{X}, \quad B[\boldsymbol{u}](t, \boldsymbol{x}) = 0, \quad (t, \boldsymbol{x}) \in [0, T] \times \partial\mathbb{X} \tag{2}$$

where $\boldsymbol{u} : [0, T] \times \mathbb{X} \to \mathbb{R}^c$ is the solution of the PDE, where $c$ is the field dimension, used to describe various field quantities such as velocity, pressure, and density, while $\boldsymbol{u}^0(x)$ is the initial condition at time $t = 0$, and $B[\boldsymbol{u}](t, \boldsymbol{x}) = 0$ are the boundary conditions at $\boldsymbol{x}$ in $\partial\mathbb{X}$, which is the boundary of the domain $\mathbb{X}$. Here, $\partial_{\boldsymbol{x}} \boldsymbol{u}, \partial_{\boldsymbol{x}\boldsymbol{x}} \boldsymbol{u}, \ldots$ are the partial derivatives of the solution $\boldsymbol{u}$ with respect to the domain, while $\partial_t \boldsymbol{u}$ is with respect to the time. The functional $F$ describes the possibly non-linear interactions between the PDE's terms.

### 2.2 PROBLEM DEFINITION

We consider PDEs (Sec. 2.1) whose solution is described as a temporal sequence of field data $\{\boldsymbol{u}^k\}_{k=0,\ldots,N} := \boldsymbol{u}^0, \boldsymbol{u}^1, \ldots, \boldsymbol{u}^N$ where $\boldsymbol{u}^k$ is the field data at time step $t_k$, that is, the state of the

physical system governed by the PDE under consideration at time $t_k$ discretized using $\Delta t = T/N$. Each $\boldsymbol{u} \in \mathcal{X} \subseteq \mathbb{R}^{c \times x_1, \dots, x_D}$ represents the field tensor data with $c$, the number of physical variables such as density and velocity, and $x_i$ the spatial dimensions of the $i$-th coordinate. For example, for a 1-d problem we have $\mathcal{X} \subseteq \mathbb{R}^{c \times x_1}$, for a 2-d problem $\mathcal{X} \subseteq \mathbb{R}^{c \times x_1 \times x_2}$, and for a 3-d problem $\mathcal{X} \subseteq \mathbb{R}^{c \times x_1 \times x_2 \times x_3}$. We will often refer to $c$ as the channel dimension. We aim to emulate numerical simulators of PDEs which iteratively map $\mathcal{M} : \mathcal{X} \to \mathcal{X}$ from $\boldsymbol{u}^k$ to $\boldsymbol{u}^{k+1}$. The emulator (or surrogate model) is a learnable function modeled as a neural network NN with weights $\boldsymbol{\theta}$. We refer to the parameters of a neural network as weights to avoid a conflict in terminology with the parameters of PDEs. In the following, we denote the emulator's prediction at time index $k$ as $\tilde{\boldsymbol{u}}^k$. Auto-regressive neural networks predict the next time step's field data based on a sequence of field data tensors of length $\ell$

$$\tilde{\boldsymbol{u}}^{k+1} = \text{NN}(\tilde{\boldsymbol{u}}^{k-\ell+1}, \dots, \tilde{\boldsymbol{u}}^k; \boldsymbol{\theta}).$$

Given the length of the input sequence $N \in \mathbb{N}$, and an initial input sequence $(\boldsymbol{u}^0, \dots, \boldsymbol{u}^{\ell-1}) = (\tilde{\boldsymbol{u}}^0, \dots, \tilde{\boldsymbol{u}}^{\ell-1})$ of length $\ell < N$, the ML model auto-regressively generates the remaining sequence $(\tilde{\boldsymbol{u}}^\ell, \dots, \tilde{\boldsymbol{u}}^N)$. The training loss is typically the normalised root-mean-square error (RMSE) between the predicted and the true field data tensors

$$\mathbf{L}(\boldsymbol{\theta}) = \sum_{k=\ell}^{N} \text{nRMSE}\left(\boldsymbol{u}^k, \ \tilde{\boldsymbol{u}}^k\right) \equiv \sum_{k=\ell}^{N} \frac{||\tilde{\boldsymbol{u}}^k - \boldsymbol{u}^k||_2}{||\boldsymbol{u}^k||_2}, \quad (3)$$

where $||u||_2$ is the $L_2$-norm of a (vector-valued) variable $u$. Since we are training an auto-regressive neural network, the gradients of the above loss can be backpropagated in time in various ways. We discuss this in the following sections. Figure 1(left) illustrates this auto-regressive approach to solving PDEs. In the vast majority of experimental setups, the assumption is made that $\ell > 1$, and, therefore, an initial input sequence of length $\ell$ is available to the model; in practice, this would require a numerical simulation to be run for $\ell - 1$-time steps from the initial condition and for each PDE parameter $\boldsymbol{\lambda}$. The main idea of CAPE is to learn to generate these sequences based on the current field data and parameter values $\boldsymbol{\lambda}$ and use those as input to an off-the-shelf neural surrogate model such as an FNO (Li et al., 2021a) or an U-Net (Ronneberger et al., 2015) to perform a complex interpolation.

### 2.3 COMBINING NEURAL PDE SOLVERS WITH THE CAPE MODULE

The proposed approach is motivated by the need for neural PDE solvers to generalize to PDE parameters unseen during training. We propose CAPE, a novel neural network architecture that takes the prior state of the system $\tilde{\boldsymbol{u}}^k$ and PDE parameters $\boldsymbol{\lambda}$ as input and predicts the $\ell$-intermediate future states $\left\{\hat{\boldsymbol{u}}_{\text{cape}}^{k \to k+i}\right\}_{i=1,\dots,\ell} = \text{CAPE}(\boldsymbol{u}^k, \boldsymbol{\lambda}; \boldsymbol{\theta}_{\text{CAPE}})$[1]. The output of CAPE is then used by the BASE network. The overall structure is provided in Fig. 1(right). The intuition behind this approach is that the intermediate future states capture information about the PDE parameters' impact by attending to the results of the convolutional operations. While we do not change the architecture of the base neural PDE solvers, we propose to use them to predict, given the past temporal states and the intermediate future states, the state for the next time step. This is contrary to the typical use of neural PDE solvers. The base network is trained jointly with the CAPE module. As shown in Sec. 3, this choice improves the prediction capability of the BASE network.

During training, the output of CAPE is regularized by the additional loss term

$$\mathbf{L}_{\text{cape}}(\boldsymbol{\theta}_{\text{CAPE}}) = \sum_{k=\ell}^{N} \sum_{i=1}^{\min(\ell, N-k)} \text{nRMSE}\left(\hat{\boldsymbol{u}}_{\text{cape}}^{k \to k+i}, \boldsymbol{u}^{k+i}\right), \quad (4)$$

which forces the CAPE module to predict a temporal sequence of future field data $\left\{\boldsymbol{u}^{k+i}\right\}_{i=1,\dots,\ell}$.

Finally, the intermediate sequence $\left\{\hat{\boldsymbol{u}}_{\text{cape}}^{k \to k+i}\right\}_{i=1,\dots,\ell}$ is concatenated with $\boldsymbol{u}^k$, the field data at time $t_k$, and given to the base network to make the final prediction. In summary, the CAPE module transforms the input variables $\{\boldsymbol{u}^k, \boldsymbol{\lambda}\}$ into temporal-sequential intermediate field data $\{\boldsymbol{u}^k, \hat{\boldsymbol{u}}_{\text{cape}}^{k \to k+1}, \dots, \hat{\boldsymbol{u}}_{\text{cape}}^{k \to k+\ell}\}$ which is then interpolated by the base neural network. Before we

---

[1]In principle it is possible for CAPE to also predict field data of past time steps: $\left\{\hat{\boldsymbol{u}}_{\text{cape}}^{k \to k+i}\right\}_{i=\pm 1,\dots,\pm\ell}$.

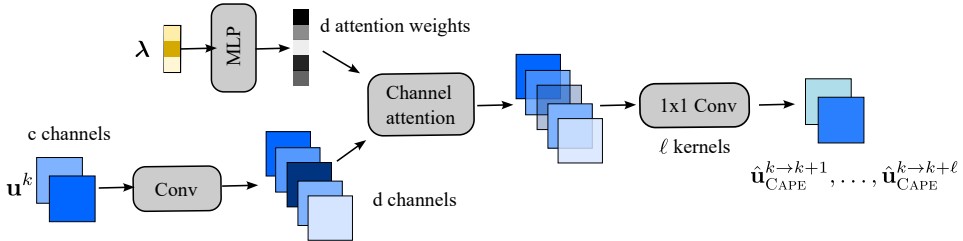

Figure 2: The CAPE module for one type of convolution (residual connections are omitted).

introduce the inductive bias of the CAPE module, we motivate the general approach from a classical numerical simulation perspective.

**CAPE as style interpolation.** Let us consider the following simple PDE:

$$\partial_t \boldsymbol{u} = F(\boldsymbol{u}; \boldsymbol{\lambda}). \tag{5}$$

The base neural network can be expressed with the equation

$$\boldsymbol{u}^{k+1} = f_{\text{base}}\left(\boldsymbol{u}^k, \left\{\hat{\boldsymbol{u}}_{\text{cape}}^{k \to k+i}\right\}_{i=1,..,\ell}; \boldsymbol{\theta}_{\text{BASE}}\right), \tag{6}$$

where $f_{\text{base}}$ is the function expressed by the base network and $\left\{\hat{\boldsymbol{u}}_{\text{cape}}^{k \to k+i}\right\}_{i=1,...,\ell} =$ CAPE$(\boldsymbol{u}^k, \boldsymbol{\lambda}; \boldsymbol{\theta}_{\text{CAPE}})$ are the intermediate future states predicted by the CAPE module generated for time index $k$. The CAPE module can be interpreted as a pre-processor network providing sequential data $\boldsymbol{u}^k, \{\boldsymbol{u}^{k+i}\}_{i=1,...,\ell}$, while Eq. 6 can be interpreted as an interpolation network applying a particular style of the given data $\boldsymbol{u}^k$ to $\{\hat{\boldsymbol{u}}_{\text{cape}}^{k+i}\}_{i=1,...,\ell}$ (see the last figure of Fig. 4).

**CAPE as implicit discretization method.** On the other hand, those equations can be understood from a numerical method perspective, where using implicit discretization (Anderson et al., 2016), Eq. 5 reduces to:

$$\boldsymbol{u}^{k+1} = \boldsymbol{u}^k + \Delta t F(\boldsymbol{u}^{k+1}; \boldsymbol{\lambda}) \equiv \tilde{F}(\boldsymbol{u}^k, \boldsymbol{u}^{k+1}; \boldsymbol{\lambda}). \tag{7}$$

Now, Eq. 6 with $\ell = 1$ can be interpreted as an approximation of the implicit method of Eq. 7. When $\ell > 1$ CAPE can therefore be seen as a generalized ML-based variant of the implicit method.

## 2.4 CHANNEL-ATTENTION-BASED PARAMETER EMBEDDING (CAPE MODULE)

CAPE computes 3 different $d$-dimensional channel attention masks $\boldsymbol{a}_\alpha \in \mathbb{R}^d, \alpha = 1, 2, 3$ from the parameters of the PDE $\boldsymbol{\lambda}$ using a 2-layer MLP

$$\boldsymbol{a}_\alpha = \boldsymbol{W}_{2,\alpha}\sigma(\boldsymbol{W}_{1,\alpha}\boldsymbol{\lambda}), \tag{8}$$

where $d$ is the channel dimension in the feature space and $\sigma$ is the GeLU activation function (Hendrycks & Gimpel, 2016). $\boldsymbol{W}_\alpha = (\boldsymbol{W}_{1,\alpha}, \boldsymbol{W}_{2,\alpha})$ are the weights associated with three operators: a $1 \times 1$-convolution ($g_1$), a depth-wise convolution ($g_2$), and a spectral convolution (Li et al., 2021a) ($g_3$), that are used to compute the tensor representations $\boldsymbol{z}_\alpha^k \in \mathbb{R}^{d \times n_x \cdots}$ as $\boldsymbol{z}_\alpha^k = g_\alpha(\boldsymbol{u}^k, \boldsymbol{W}_\alpha)$. The tensors are then multiplied by the attention

$$\boldsymbol{v}_\alpha^k = \boldsymbol{a}_\alpha^k \odot_1 \boldsymbol{z}_\alpha^k \tag{9}$$

using the Hadamard operator ($\odot_1$) over the first dimension (the channel dimension) which is equivalent to the broadcast operation of ML programming languages. The three convolutions can be interpreted as a finite difference method since convolution operations accumulate local information of a mesh, which, in principle, can simulate local interactions such as advection and diffusion. Intuitively, channel attention is equivalent to choosing an appropriate physical process for each PDE parameter. A similar mechanism has been proposed for visual tasks, called the squeeze-and-excitation networks (Hu et al., 2018) which enhances useful channels of the feature vector of convolutional networks through an attention mechanism. The feature $\boldsymbol{v}_\alpha^k \in \mathbb{R}^{d \times n_x \cdots}, \alpha = 1, , 2, 3$ are combined to form an intermediate feature $\boldsymbol{y}^k \in \mathbb{R}^{c \times \ell \times n_x \cdots}$ as

$$\boldsymbol{y}^k = h_{1 \times 1, d \to c \times \ell}\left(\sigma\left(h_{1 \times 1, c \to d}(\boldsymbol{u}^k) + \sum_\alpha \boldsymbol{v}_\alpha^k\right)\right) \tag{10}$$

| PDE | training parameters | testing (unseen) parameters |
|---|---|---|
| 1D Advection | $\beta = (0.2, 0.4, 0.7, 2.0, 4.0)$ | $\beta = (0.1, 1.0, 7.0)$ |
| 1D Burgers | $\nu = (0.002, 0.007, 0.02, 0.04, 0.2, 0.4, 2.0)$ | $\nu = (0.001, 0.01, 0.1, 1.0, 4.0)$ |
| 2D NS | $\eta = \zeta = (10^{-8}, 0.001, 0.004, 0.01, 0.04, 0.1)$ | $\eta = \zeta = (0.007, 0.07)$ |

Table 1: PDE parameters used in the experiments.

where $h_{1\times1,*}$ are $1 \times 1$ convolutions that adjust the number of dimensions, in particular $h_{1\times1,c\to d}$ : $c \times n_x \cdots \to d \times n_x \ldots$, while $h_{1\times1,d\to c\times\ell} : d \times n_x \cdots \to c \times \ell \times n_x \ldots$. Finally, the sequence of predictions is computed

$$\left\{\boldsymbol{u}_{\text{cape}}^{k\to k+i}\right\}_{i=1,\ldots,\ell} = (\boldsymbol{u}^k + \text{LayerNorm}(\boldsymbol{y}_i^k))_{i=1,\ldots,\ell} \qquad (11)$$

where $\boldsymbol{y}_i^k$ is the $i$-th element of the data tensor $\boldsymbol{y}^k$, selected from the second dimension [2]. For the sake of presentation, we omitted the batch dimension. Figure 2 illustrates the architecture of the CAPE module.

## 2.5 CURRICULUM LEARNING

For each initial condition $\boldsymbol{u}^0$, the BASE and CAPE models (referred to as $\text{NN}(\boldsymbol{u}^k; \boldsymbol{\theta})$) jointly predict the full temporal sequence $\{\boldsymbol{u}^k\}_{k=1,\ldots,N}$. We propose the following *curriculum learning* strategy. For each training epoch, we split the temporal sequence into two parts. For the first part $(\boldsymbol{u}^0, \ldots, \boldsymbol{u}^{k_{\text{trans}}})$ we use *auto-regressive* training using $\tilde{\boldsymbol{u}}^k$, the prediction of the model at time index $k$, as input to predict the solution for time index $k+1$, that is, $\tilde{\boldsymbol{u}}^{k+1} = \text{NN}(\tilde{\boldsymbol{u}}^k; \boldsymbol{\theta})$. For the second part $(\boldsymbol{u}^{k_{\text{trans}}+1}, \ldots, \boldsymbol{u}^N)$, we train the model using *teacher-forcing*. The teacher-forcing strategy (Williams & Zipser, 1989; Bengio et al., 2015) computes the prediction for time index $k+1$ using a noisy version of the true value at time index $k$, that is, $\tilde{\boldsymbol{u}}^{k+1} = \text{NN}(\boldsymbol{u}^k + \epsilon; \boldsymbol{\theta})$ where $\epsilon$ is random noise increasing the stability at inference time (Sanchez-Gonzalez et al., 2018; 2020; Pfaff et al., 2020; Stachenfeld et al., 2021). The time index $k_{\text{trans}}$ determines the time step where we switch from auto-regressive training to teacher-forcing and is computed using the following monotonically increasing function of the epoch number $n$

$$k_{\text{trans}} = \left\lfloor \frac{N}{2} \left(1 + \tanh\left[\frac{\frac{n}{N} - 0.5}{\Delta}\right]\right) \right\rfloor, \qquad (12)$$

where $N$ is the total epoch number, and $\Delta$ is a hyper-parameter controlling the steepness of the transition function. A plot of the function is provided in Fig. 6 and a detailed algorithm of the training strategy is provided in Appendix (Algorithm 1).

The strategy is based on the following two assumptions: (1) the prediction error decreases as the number of training epochs increases, (2) the accumulated error increases as the number of auto-regressive rollout steps increases. Teacher-forcing training is usually more stable since it avoids the accumulation of prediction errors and should be used exclusively in the first phase of training. The auto-regressive strategy simulates the behavior at test time and exposes the model to inputs that evolved further from the true data, making it more robust to error accumulation. For the same reasons, however, it tends to be less stable, especially in the early phase of training. The proposed curriculum-learning strategy is used to combine the advantages of both approaches.

## 3 EXPERIMENTS

We used datasets provided by PDEBench (Takamoto et al., 2022) a benchmark for SciML from which we selected the following PDEs:

**1D Advection Equation.** This equation describes the pure-advection of waves

$$\partial_t u(t, x) + \beta \partial_x u(t, x) = 0, \qquad (13)$$

---

[2]We found that in the case of 2D NS we obtain improved results when modifying the right-hand side of Eq. 11 as: $\boldsymbol{u}^k(1 + \text{LN}(\boldsymbol{y}_i^k))$ and we applied it to obtain the 2D NS results in this paper.

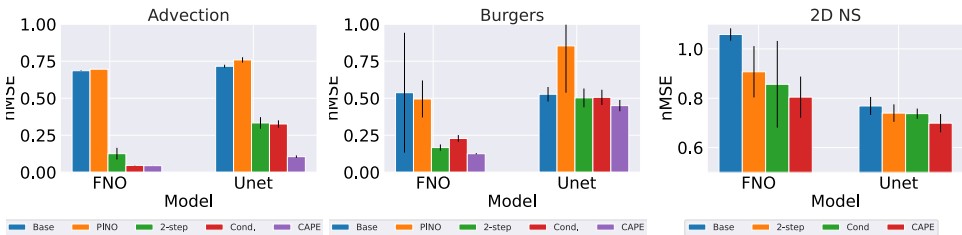

Figure 3: Plots of the normalized MSE (smaller is better) with an error bar for Advection eq. (Left), Burgers eq. (Middle), and 2D Compressible NS equations (Right).

where $\beta$ is the PDE parameter describing advection velocity. The exact solution of this equation is: $u(t, x) = u_0(t, x - \beta t)$ where $u_0$ is the initial condition. Hence, this PDE can be used to check if the ML models understand the property of advection, updating the solution by just advecting the initial profile without changing it.

**1D Burgers Equation.** Burgers' equation is a mathematical model equation simulating the non-linearity and diffusivity in the hydrodynamic equation by a scalar variable

$$\partial_t u(t, x) + u(t, x)\partial_x u(t, x) = \nu/\pi\partial_{xx}u(t, x), \qquad (14)$$

where $\nu$ is the diffusion coefficient and the parameter of this equation. This PDE can be used to check if ML models can understand the non-linear behavior from the second term in the left-hand side of Eq. 14 and the diffusion process whose strength is controlled by the parameter $\nu$.

**2D Compressible Navier-Stokes Equations (2D NS).** The compressible Navier-Stokes equations (NS eqs., in the following) is one of the basic physics equations describing classical fluid dynamics

$$\partial_t \rho + \nabla \cdot (\rho\mathbf{v}) = 0, \quad \rho(\partial_t\mathbf{v} + \mathbf{v} \cdot \nabla\mathbf{v}) = -\nabla p + \eta\triangle\mathbf{v} + (\zeta + \eta/3)\nabla(\nabla \cdot \mathbf{v}), \qquad (15)$$

$$\partial_t \left[\epsilon + \frac{\rho v^2}{2}\right] + \nabla \cdot \left[\left(\epsilon + p + \frac{\rho v^2}{2}\right)\mathbf{v} - \mathbf{v} \cdot \sigma'\right] = 0, \qquad (16)$$

where $\rho$ is the mass density, $\mathbf{v}$ is the velocity, $p$ is the gas pressure, $\epsilon = p/(\Gamma - 1)$ is the internal energy, $\Gamma = 5/3$, $\sigma'$ is the viscous stress tensor, and $\eta, \zeta$ are the shear and bulk viscosity, respectively. For 1-dimensional PDEs, we used $N = 9000$ training instances and 1000 test instances for each PDE parameter with resolution 128. For 2-dimensional NS equations, we used $N = 900$ training instances and 100 test instances for each PDE parameter with spatial resolution $64 \times 64$.

**Experiment Setup.** We evaluated the neural models U-Net (Ronneberger et al., 2015) and FNO (Li et al., 2021a) with some datasets provided by PDEBench (Takamoto et al., 2022) for various parameters for the 1D Advection equation, 1D Burgers equation, and 2D compressible Navier-Stokes equations. We also evaluated the message passing neural PDE Solvers (MPNN) (Brandstetter et al., 2022) as a baseline allowing conditional treatment of PDE parameters. We trained each of the neural models (1) **Base**: without any changes (vanilla model), (2) **PINO**: with a PINO loss (Li et al., 2021b), (3) **Conditional**: the parameters are added to the input data as new channel-dimensions, (4) **2-step**: with the field data for the current and previous time-steps as input ($\boldsymbol{u}^k, \boldsymbol{u}^{k-1}$), and (5) **CAPE**: with the CAPE module. Other than case (4), we only provided field data for one time step to the models and, therefore, the models cannot obtain PDE parameters' information from the given data. The PINO loss function regularizes the ML models to satisfy the residuals of the PDEs and might lead to an improved generalization behavior for unseen PDE parameters. The CAPE module predicts intermediate field data for one future time step and this is used as the input to the BASE model together with the field data of the current time step.[3]. The amount of field data provided to the BASE network in cases (4) and (5) is the same: in case (4) the model always obtains field data for time steps $k$ and $k-1$ as input to predict the field data for time step $k+1$ while in case (5) the BASE model obtains field data for time step $k$ and intermediate field data for time step $k+1$ generated by the CAPE model to predict the field data for time step $k+1$. Hence, the model in case (4) obtains two true field data for 2 initial steps and, therefore, has the opportunity to adapt to different PDE parameter values. Hence, while case (4) is more expensive, we consider it a strong baseline for the problem.

---

[3]First introduced in (Li et al., 2021a) with twenty steps as input.

Figure 4: Visualization of the results: Advection eq. at the final time-step ($t = 2.0$) (Left), Burgers eq. at $t_k = 20(t = 1.0)$ (2nd-left) at the final time-step, and $V_x$ of 2D NS equations at $t_k = 5(t = 0.25)$ (Right). Here "Base" is the vanilla FNO, "CAPE " is the FNO with CAPE, "CAPE module" is the direct output from CAPE module only; the CAPE module provides a higher frequency proposal to the BASE model which then more accurately predicts the field data.

| PDE | model | BASE | BASE (PINO) | Conditional | prev. 2-steps | CAPE |
|-----|-------|------|-------------|-------------|---------------|------|
| Advection | FNO | $0.69^{\pm 2.2 \times 10^{-3}}$ | $0.70^{\pm 1.6 \times 10^{-4}}$ | $0.05^{\pm 1.2 \times 10^{-3}}$ | $0.13^{\pm 3.9 \times 10^{-2}}$ | $\mathbf{0.04}^{\pm 3.2 \times 10^{-4}}$ |
| | Unet | $0.72^{\pm 1.0 \times 10^{-2}}$ | $0.76^{\pm 1.8 \times 10^{-2}}$ | $0.33^{\pm 2.0 \times 10^{-2}}$ | $0.33^{\pm 3.9 \times 10^{-2}}$ | $\mathbf{0.11}^{\pm 8.3 \times 10^{-3}}$ |
| | MPNN | $0.33^{\pm 2.5 \times 10^{-2}}$ | – | $0.34^{\pm 1.7 \times 10^{-2}}$ | – | – |
| Burgers | FNO | $0.54^{\pm 0.40}$ | $0.49^{\pm 1.3 \times 10^{-1}}$ | $0.23^{\pm 2.3 \times 10^{-2}}$ | $0.17^{\pm 2.1 \times 10^{-2}}$ | $\mathbf{0.13}^{\pm 4.4 \times 10^{-3}}$ |
| | Unet | $0.53^{\pm 4.9 \times 10^{-2}}$ | $0.85^{\pm 3.2 \times 10^{-1}}$ | $0.51^{\pm 5.1 \times 10^{-2}}$ | $0.50^{\pm 6.3 \times 10^{-2}}$ | $\mathbf{0.45}^{\pm 3.8 \times 10^{-2}}$ |
| | MPNN | $0.26$ | – | $0.28$ | – | – |
| 2D NS | FNO | $1.06^{\pm 2.5 \times 10^{-2}}$ | – | $0.86^{\pm 0.18}$ | $0.91^{\pm 0.10}$ | $\mathbf{0.80}^{\pm 8.3 \times 10^{-2}}$ |
| | Unet | $0.77^{\pm 3.6 \times 10^{-2}}$ | – | $0.74^{\pm 0.02}$ | $0.74^{\pm 3.5 \times 10^{-2}}$ | $\mathbf{0.70}^{\pm 3.7 \times 10^{-2}}$ |

Table 2: List of the normalized RMSE (the smaller, the better) for Advection eq., Burgers eq., and 2D Compressible NS equations.

Since the solutions of each PDE are not normalized and based on prior results on evaluating PDE solvers Takamoto et al. (2022), we measure the normalized RMSE (nRMSE). We used the normalized RMSE loss function $L_{\mathrm{nRMSE}}$ with the auxiliary loss function of the CAPE module $\mathbf{L}_{\mathrm{cape}} := \mathbf{L}_{\mathrm{nMSE}} + \alpha \mathbf{L}_{\mathrm{cape}}$ where $\alpha$ is the weight coefficient. The optimization was performed with Adam (Kingma & Ba) for 100 epochs. The learning rate was divided by 2.0 every 20 epochs. For a fair comparison, we made the model size of the different methods as similar as possible. A table with parameter sizes is provided in Tab. 6 in the appendix. A more detailed description of the hyper-parameters is provided in Appendix B.

**Varying the parameter values.** Fig. 3 shows bar plots comparing the BASE models with and without CAPE module, the models with PINO loss, and the models with the 2-steps as input. The CAPE module results in the lowest error in all cases. In particular, the CAPE module leads to an impressible error reduction ranging from 20 % (2D NS equation) to 95 % (1D Advection). We partly attribute this to the BASE network's ability to capture physical dynamics from the PDE parameter-dependent data provided by the CAPE module. The vanilla FNO is a state-of-the-art model and is superior to the U-net as a BASE model [4]. Interestingly, the PINO loss provides almost no benefit in our setting. We hypothesize that the PINO loss is heavily affected by and dependent on the time-step size. A more detailed explanation of this observation is given in Appendix C. Interestingly, the CAPE module provides either comparable or a little better results than the case with 2-step information. This indicates that the CAPE module succeeded in providing equivalent and even more useful information to the BASE network.

**Generalization Ability.** Fig. 5 plots the normalized MSE for each parameter value of the PDEs using FNO as the BASE network. The parameter of the 1D Advection PDE controls the advection velocity and the parameters of the remaining equations control the strength of the diffusion process.

---

[4]For the 2D NS PDE, the U-net achieves a smaller error than the FNO. We hypothesize that this is partly due to the difference in model size (the number of weights of the U-net is nearly 10 times larger) and partly because the U-net typically excels at image-to-image mapping problems.

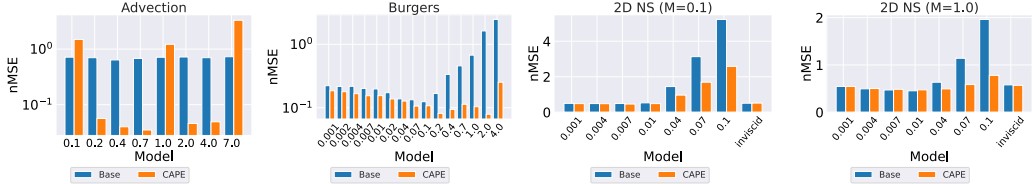

Figure 5: Plots of the normalized MSE (smaller is better) in terms of each PDE parameter for Advection eq. (Left), Burgers eq. (Middle-Left), 2D NS with $M = 0.1$ (Middle-right), and 2D NS with $M = 1$ (Right). The results were obtained using the FNO.

| PDE | Model | Ablation | nMSE | |
|---|---|---|---|---|
| 2D NS | FNO | curriculum strategy | $\mathbf{8.0 \times 10^{-1}}$ | |
| | | pure Autoregressive | $1.3 \times 10^{+0}$ | $(+0.5)$ |
| | | pure Teacher-Forcing | $3.2 \times 10^{+0}$ | $(+\mathbf{2.4})$ |
| 2D NS | Unet | curriculum strategy | $\mathbf{7.0 \times 10^{-1}}$ | |
| | | pure Autoregressive | $1.0 \times 10^{+0}$ | $(+\mathbf{0.3})$ |
| | | pure Teacher-Forcing | $1.0 \times 10^{+0}$ | $(+\mathbf{0.3})$ |

Table 3: Ablation study for the 2D CFD equations with FNO and Unet as BASE model.

First, the 1D-Advection result shows that the CAPE module overfits with the trained parameter ($\beta = 0.2, 0.4, 0.7, 2.0, 4.0$), though it showed a very nice generalization performance to the trained PDE parameters. This could also be indicated by the fact that only the CAPE result for the 1D Advection equation showed a much lower error than the approach receiving the 2-step field data which in theory provides a similar amount of information as the CAPE module. Hence, to evaluate generalization to the unseen PDE parameters, we expect the difference of these errors to be correlated with the CAPE module overfitting to the trained parameters.

On the other hand, the other cases (the parameter describes the diffusion process) showed a good generalization to unseen PDE parameters. Note that the plots also indicate that vanilla models show a preference for the parameter regime; in all the cases, the vanilla models exhibit better results on smaller diffusion coefficients but lose accuracy as the diffusion coefficients increase.

**Ablation experiments.** In this section, we performed an ablation study to separate the impact of the curriculum learning strategy. The ablation study on CAPE structure is provided in Tab. 9 in Appendix. Tab. 3 lists the result for the 2D NS equations using FNO and Unet as the BASE network. The proposed curriculum learning strategy drastically impacts the accuracy of the model in all cases, indicating the effectiveness of seamlessly bridging teacher-forcing and auto-regressive training. The full ablation study results are provided in Tab. 8 in Appendix.

**Qualitative analysis of the CAPE module.** In Fig. 4 we plot some representative outputs of the vanilla FNO, the CAPE module, and the overall CAPE model, and compare them with the true solutions. Interestingly, we can see that the BASE network often interpolates a higher noise approximation of the CAPE module into the typical shape (style) of the final solution.

**Inference time.** We provide the comparison of the inference time between the hybrid approach (including 2 initial steps as input as done in the FNO paper) and CAPE (only using the initial step as the input). Here we only consider a scenario where the initial time steps are obtained from a numerical simulation. Tab. 4 lists the inference time in the case of the 2D NS equation. The inference time of the FNO with the CAPE module is much shorter than the hybrid method where the inference time is dominated by the simulation time [5].

---

[5]The experiments were run using an Nvidia GeForce RTX 3090 with CUDA-11.6. The ML models are implemented using PyTorch 1.12.1 and the numerical simulations with JAX-0.3.17.

| PDE | Resolution | Total Inference time [sec] | simulation time [sec] |
|---|---|---|---|
| Simulation + FNO | $512^2$ | 582.8 | 582.6 |
| CAPE + FNO | $512^2$ | 1.3 | – |

Table 4: Inference time comparison of simulation (initial steps=10) + FNO and CAPE + FNO in the case of 2D CFD ($\eta = \zeta = 0.1$). The time-step size is $\Delta t = 0.05$ and the computations were performed until $t = 1.0$ as in this paper's other experiments.

## 4 RELATED WORK

**Scientific Machine Learning Models**   Scientific Machine Learning aims at data-driven modeling of physical systems. A notable example is Physics Informed Neural Networks (Cai et al., 2022) (PINNs) that, having access to the PDE of a system, learns a neural network over the domain $\mathcal{X} \to \mathbb{R}^d$ by enforcing small residual, i.e. the error when the solution is evaluated by the PDE or the boundary conditions, over a set of sampled points. While PINNs have the capacity to model various physical systems, they need to be trained for each new condition or parameter. Neural Operators (Li et al., 2021b), as FNO (Li et al., 2021a) or Graph NO (Li et al., 2020), models the continuous operators over an infinite space and have shown the ability to generalization at multiple scales. Also, more traditional image-to-image neural networks such as the U-Net (Ronneberger et al., 2015) can be adopted to model NOs. Physics-Informed Neural Operators (PINO) (Li et al., 2021b) improve the representational power of PINNs by pre-training a NO but having similar limitations to NOs. Message passing neural PDE solver (Brandstetter et al., 2022) extends the message passing principle to solve PDEs. In signal processing, for the artificial bandwidth extension (ABE) task, the Time-Frequency Network(Dong et al., 2020) (TFNet) has been proposed, which shares a similar concept of channel attention (CA).

**Training Autoregressive Models**   As was discussed in Sec. 2.5, there are two representative training strategies for Scientific ML, that is, teacher-forcing and auto-regressive training. The teacher-forcing strategy was originally developed in natural language processing (Williams & Zipser, 1989; Bengio et al., 2015) which predicts n+1-th step data (word) using the true n-th step information (word). This method is known to prevent from the error-accumulation in the predicted sequential data during the model training. In the case of Scientific ML, it was found that it is profitable to add a random noise for improving the robustness against the accumulated error at the inference time (Sanchez-Gonzalez et al., 2018; 2020; Pfaff et al., 2020; Stachenfeld et al., 2021). On the other hand, the auto-regressive strategy uses the previous prediction of the model as n-th timestep information. Because of the error accumulation problem, not so many works were adopted, e.g. (Li et al., 2021a). Recently, (Brandstetter et al., 2022) proposed a reconciling method for this problem which is the so-called "pushforward trick". To increase stability, this method uses an adversarial-style loss which predicts the next timestep data using the previous prediction which is calculated using true data. Note that our method adopted a curriculum training strategy to prevent error accumulation instead of using an additional loss function.

## 5 CONCLUSION

This paper proposed a CAPE module which allows any data-driven SciML models to incorporate PDE parameters. We also proposed a simple but effective curriculum training strategy that allows us to bridge teacher-forcing and auto-regressive learning. We performed an extensive set of experiments and showed the effectiveness and efficiency of our method from various aspects: generalization of seen/unseen PDE parameters during training, parameter efficiency, and inference time.

One of our key findings is the behavior of ML models without parameter embedding. We found that the ML models without parameter embeddings either (1) show a poor performance uniformly for all the PDE parameters (1D Advection eq.), (2) overfitting to a preferred parameter regime (1D Burgers eq. and 2D NS eqs.). Hence, the ML model in this case has no generalization power in terms of PDE parameters. On the other hand, the models with CAPE module generalized well for 1D Burgers' eq. and 2D NS eqs. whose parameters govern the physical systems' diffusion behavior. CAPE cannot be generalized for 1D Advection eq., and we consider it necessary to formulate CAPE to be more physics-informed which we consider for future work.

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
