# OpenReview forum: "CAPE: Channel-Attention-Based PDE Parameter Embeddings for SciML"
_ICLR.cc/2023/Conference — Submitted to ICLR 2023_

### Official Review · Reviewer_PGvV · 2022-10-23

**Confidence:** 3
**Correctness:** 2
**Technical Novelty And Significance:** 2
**Empirical Novelty And Significance:** 2
**Recommendation:** 6

**Clarity, Quality, Novelty And Reproducibility:**

Quality: medium

Clarity: fair

Novelty: fair

Reproducibility: fair.

**Strength And Weaknesses:**

Strengths:

The paper addresses an important problem. The method seems to be novel.

Weaknesses:

It is not well justified or motivated why introduce this module. And why the simplest method of putting the parameter (or a transformation of the parameter) does not work. The improved accuracy of the module can be due to:
(a) The CAPE module contains the information of the PDE parameter
(b) It has more parameters than the model without this module, thus have higher expressivity
(c) The CAPE module needs to predict more future steps, thus having a regularizing effect.
(d) The inductive bias of the CAPE module is useful and can generalize better

I believe that the simple method of providing the PDE parameter will also benefit from the reason (a). To really demonstrate the usefulness of the CAPE module, the paper needs to rule out (a)(b)(c) (and other potential reasons) and then (d) can then be the likely reason.

To rule out (a), I think at least 2 more baseline are necessary:
(I) a base network which takes as input the concatenation of the field data with the PDE parameter (expanded into channel features) concatenated on the channel dimension, i.e.
u^{k+1} = Base([u^k,\lambda])

(II) a base network which takes as input the concatenation of the field data with an embedding of PDE parameter:
u^{k+1} = Base([u^k,MLP(\lambda)])
where the MLP can be a 2- or 3-layer simple neural network. The layer number and the number of neurons needs to be hyperparameter searched.

In my previous experience, the above two simple methods works quite well, and can be a strong baseline.

To rule out (b), add another baseline,
(III) a larger base network whose number of parameter is similar to the (4) CAPE + base network.

To rule out (c), the method needs to compare with a baseline where the base network needs to predict the same number of future steps as the CAPE.

The above baselines need to be hyperparameter searched with similar budget as in CAPE, to ensure a fair comparison.

There can also be other potential factors, but I think the above are necessary to show the benefit of CAPE.

Also, the CAPE architecture needs to be better justified

--
Update:
Through the rebuttal, the reviewer has addressed my concerns and resolved the misunderstanding. Thus I have increased my score.


**Summary Of The Paper:**

This paper proposes a channel-attention-based parameter embedding (CAPE) component for PDEs. The module takes as input the state time step t, and the PDE parameter labmda, predicts the state for several intermediate steps, which the main network takes as input (together with state at time t) to predict state at t+1. It details the architecture of the module, and also show in experiments that it results in improved accuracy.

**Summary Of The Review:**

In summary, the module the paper proposes is interesting. However, in its current form, there is no enough empirical evidence to justify the benefits of CAPE. Therefore, in its current form, I recommend for reject. On the other hand, if the concerns are addressed, I'm willing to increase my score.

---

> ### Author Response · Authors · 2022-11-14
> **Reply to Reviewer PGvV**
>
> Dear Reviewer PGvV,
>
> Thank you very much for your kind comment for improving our paper. The following are our responses.
>
> >”It is not well justified or motivated why introduce this module. “
>
> Thank you very much for your kindly pointing out. In the introduction, we have expressed the limitation of current approaches (second paragraph) as:
> “The majority of these methods,[...]”,
>
> Note that this is also justified by the performance of the proposed method with respect to existing models, table 2, for example.
>
> >”And why the simplest method of putting the parameter (or a transformation of the parameter) does not work. The improved accuracy of the module can be due to: (a) The CAPE module contains the information of the PDE parameter (b) It has more parameters than the model without this module, thus have higher expressivity (c) The CAPE module needs to predict more future steps, thus having a regularizing effect. (d) The inductive bias of the CAPE module is useful and can generalize better”
>
> Thank you for the comment, we added the ablation study to highlight which components contribute to the final performance. We show that point (d) is happening and is based on the capability (c) of CAPE to predict multiple timesteps. Contrary to the Reviewer’s mentions, it is not possible to regularize when generating future time steps, since otherwise, the CAPE module will not be able to help the Base model to improve the prediction. About point (a) this is necessary to distinguish which parameter is active. We added the comparison with other baselines in the new version of the paper.
> Again we welcome any additional clarifying feedback from you on this matter.
>
> >”I believe that the simple method of providing the PDE parameter will also benefit from the reason (a). To really demonstrate the usefulness of the CAPE module, the paper needs to rule out (a)(b)(c) (and other potential reasons) and then (d) can then be the likely reason.
> To rule out (a), I think at least 2 more baseline are necessary: (I) a base network which takes as input the concatenation of the field data with the PDE parameter (expanded into channel features) concatenated on the channel dimension, i.e. u^{k+1} = Base([u^k,\lambda])
> (II) a base network which takes as input the concatenation of the field data with an embedding of PDE parameter: u^{k+1} = Base([u^k,MLP(\lambda)]) where the MLP can be a 2- or 3-layer simple neural network. The layer number and the number of neurons needs to be hyperparameter searched.... (We are sorry for our omitting the remaining part due to length limitation)
>
> Thank you for your very kind suggestions to improve our paper. Following your advice, we have added the results for a conditional model (case (I)). A generic explanation is provided in the reply to all the reviewers.
>
> Concerning case (II), u^{k+1} = Base([u^k,MLP(\lambda)]): This idea is model-dependent to realize because all the intermediate states have spatial dimensions and neither the FNO nor the UNet perform global average pooling inside of the networks. There is a large number of possible ways to include MLP(\lambda) and this would heavily depend on the base network. The aim of our paper was a method that could be combined with any base network without modifying its structure. For this reason, we did not explore case (II). Again, this would require exploring a large search space of the model's structures specific to the particular base networks.
>
> Concerning your comment on ruling out confounding reasons for CAPE performing well.
> Case (b) is already ruled-out in the paper because the number of model parameters is set nearly the same in all the experiments, as explained in the original submission in section 3 and table 6 of the appendix.
> Case (a) is ruled out by the newly added experiments following your advice. They show that  CAPE works better than a simple conditional model with, on average, a 30% error reduction.
> Regarding case (c), the experiments showed only one-timestep future prediction is enough, and regularization effects would be quite small (though we may misunderstand your intentions).
> Again we welcome any additional clarifying feedback from you on this matter.
>
> >”Also, the CAPE architecture needs to be better justified”
>
> We have modified a part of the “Introduction” as:
>
> “In this paper, we propose a new and effective parameter embedding module by utilizing the channel-attention method inspired by numerical solver with the implicit method and style transfer in ML.”
>
> We also emphasize that motivation of the CAPE module structure was provided in section 2.3 of the original submission, in particular, in subsections “CAPE as implicit discretization method” and “CAPE as style interpolation”. Concerning the possible structure of the CAPE module, we also added an ablation study on CAPE’s structure in section F and Table 9 in Appendix.

---

> > ### Comment · Reviewer_PGvV · 2022-12-06
> > **Response**
> >
> > Thanks very much for the additional experiment which makes the paper better. From the new additional experiment (Conditional), we see that it significantly outperforms other baselines, and only slightly underperforms the proposed CAPE method (e.g. in 2D NS and Advection with FNO). It may not be convincing CAPE significantly outperforms the simple baseline to justify the added complexity.
> >
> > What I suggest by ruling out (c) is as follows: in the surrogate model training, if we train with 1-step loss, a good 1-step loss may not necessarily lead to a small multi-step loss with autoregressive rollout, since the model may overfit to the 1-step target, and be prone to the deviation during autoregressive rollout where the input (with model's prediction at the previous time step) deviates from the distribution of the training data. Typically, if we train with autoregressive rollout, and minimize the multiple-step's loss together (1-step, 2-step, 3-step loss ), it will significantly improve performance. Therefore, one reason why CAPE may work better is that the CAPE module is trained with multiple time step's loss. Therefore, a simple baseline to compare with is as follows: Take the "Conditional" module, and during training, perform multiple-step's of rollout (same number of step as CAPE module), and minimize the loss of:
> >
> > loss = 1-step-loss + 2-step-loss + ... + n-step-loss.
> >
> > If this baseline still underperforms the CAPE module, then I will be convinced that CAPE's improvement is really due to its inductive bias.
> >
> > Thanks

---

> > > ### Author Response · Authors · 2022-12-06
> > > **Reply**
> > >
> > > Thank you very much for your kind response and comments.
> > > We really appreciate the reviewer's kind reaction during this busy time.
> > > However, we would like to express our disappointment with the comments on the following points.
> > >
> > > Firstly, the reviewer's kind comment (section 2) demands us additional experiments which we are not allowed to do at this point. We hoped the suggestions were provided before the deadline for the revision (Nov. 18th). In addition, in our experiments, CAPE module only predict 1 time step future (the same as the main network and the other methods), so we would like to claim that the proposed mechanics does not contribute to CAPE module, at least in our present setting.
> > >
> > > Secondly, although the reviewer commented to us that CAPE does not significantly outperform the conditional method, we feel the comment is not fair from an objective viewpoint because the reviewer does not provided us with any concrete threshold value. We would like to emphasize that CAPE model still outperforms the conditional method by 5 to 10 % margin in most cases (note that our evaluation metric is normalized), which we think is not a negligible margin in general.
> > >
> > > Thirdly, the reviewer commented:
> > > >It may not be convincing CAPE significantly outperforms the simple baseline to justify the added complexity.
> > >
> > > We consider that this is not logical because the parameter number is the same between the conditional case and CAPE case; Besides, as is well-known, "complexity" is not a well-defined variable in DNN community, and also it does not mean the more complex model is the better one. Moreover, we cannot compare CAPE with non-existing "more complex models" (note that we have already performed a comparison with "non pre-existed conditional method").
> > >
> > > Again we really appreciate the reviewer's kind response!

---

> > > > ### Comment · Reviewer_PGvV · 2022-12-13
> > > > **Response**
> > > >
> > > > Although the main network of CAPE is one-step as other baselines, the additional input u_cape^{k->k+i}, i=1,2,...l does predict multiple time steps into the future and provides that information to the main network as input. Therefore, the most convincing baseline would be a simple baseline with the "conditional" baseline, but then during training time, also rollout l steps (the same as the l in u_cape^{k->k+l}) into the future, and the loss is the summation of
> > > >
> > > > loss = 1-step-loss + 2-step-loss + ... + l-step-loss.
> > > >
> > > > If CAPE still outperforms this baseline, it will eliminate the possibility that the improved performance is due to that CAPE is trained with more steps of rollout (as in Eq. 4), because as we know, training with more steps of rollout will significantly improve performance.
> > > >
> > > > Giving the lack of this convincing result, I keep my score. Nevertheless, this is an interesting work and if done in a more solid way, can be a good contribution.

---

> > > > > ### Author Response · Authors · 2022-12-13
> > > > > **Reply**
> > > > >
> > > > > We really appreciate your kind reply even at the last minute.
> > > > > We consider that we have understood the point of the misunderstanding each other.
> > > > > The followings are the summary of our points, hopefully enough to convince you:
> > > > >
> > > > > (1) First of all, the CAPE module's output:\hat{u_cape}^{k->k+i} is also only one-step future in our experiment (i=1), as explained in Page 6's last 5 sentences (Sec 3, Experiment Setup.). So, the input to the main network is {u^k, \hat{u}^{k+1}_CAPE}, which means that in our experiment the effect you kindly explained to us did not contribute to our result. Theoretically, it is possible to perform with l > 1, but we have not observed impressive enhancement for the moment.
> > > > >
> > > > > (2) As is presented in the Table in "General Reply to All the reviewers" and in the paper, our CAPE module is better or at least comparable to all the baseline models (conditional, prev.2-steps, MPNN). In particular, MPNN is the present state-of-the-art model for taking into account various PDE parameters, and we again would like to emphasize that this fact itself showed strong evidence of our CAPE module's usefulness.
> > > > > (Note that the parameter number of each model is:  FNO: 73k, FNO w.t. CAPE: 68k, MPNN: 641k, Unet: 2.71M, Unet w.t. CAPE: 2.75M, so that FNO w.t. CAPE is better than MPNN even with smaller parameter size.)

---

> > > > > > ### Comment · Reviewer_PGvV · 2022-12-13
> > > > > > **Response**
> > > > > >
> > > > > > Thanks the authors for addressing this misunderstanding. In light of the resolved concerns, I have increased my score

---

> > > > > > > ### Author Response · Authors · 2022-12-13
> > > > > > > **Reply**
> > > > > > >
> > > > > > > We appreciate your patience and very kind suggestions!
> > > > > > > We really enjoyed these very fruitful discussions.

---

### Official Review · Reviewer_adWD · 2022-11-03

**Confidence:** 4
**Clarity, Quality, Novelty And Reproducibility:** See above
**Correctness:** 3
**Technical Novelty And Significance:** 2
**Empirical Novelty And Significance:** 2
**Recommendation:** 3

**Strength And Weaknesses:**

Strength:

* The proposed method is easy to understand, and the writing is clear.

* The proposed method can reduce inference time since the method alleviates the need of running simulations.


Weaknesses:

* My main concerns are about experiments.

  * The authors mentioned in the introduction that several approaches can be used to incorporate the PDE parameters into the model, such as adding the PDE parameters as additional inputs, and include the parameters in the embedding module. However, it is not clear why these methods are not good. And the authors should compare with these methods in the experiments.

  * The proposed method contains more parameters than existing methods because of the introduction of the channel attention module. Therefore, the current comparison is not fair. The authors should train all models with the same number of FLOPs (or equivalently same training time).

* I also have concerns about the proposed curriculum learning.

  * The design of the proposed curriculum learning seems rather arbitrary. What is the intuition here? Especially for Eq. 12, I cannot see the benefit of this particular deign. The authors should elaborate more on this part.
  * From the results in Table 3, it doesn’t seem like the curriculum learning method is effective. For the advection equation, w/o curriculum learning has the best performance. And for the Burgers equation, the curriculum learning method also does not increase model performance.

Update:

I still believe the paper is below the acceptance bar. I've decided to keep my rating.

**Summary Of The Paper:**

This paper proposed CAPE, which is a channel attention module that embeds PDE parameters. The module can be combined with off-the-shelf PDE solver. CAPE facilitates models to adapt to unseen PDE parameters, and is beneficial for inference efficiency. The authors also propose a curriculum learning strategy to improve model performance. Experiments are provided to demonstrate the effectiveness of the proposed method.

**Summary Of The Review:**

See above

---

> ### Author Response · Authors · 2022-11-14
> **Reply to Reviewer adWD**
>
> Dear Reviewer adWD,
>
> Thank you  for your kind comments and helping to improve our paper. The following are our responses.
>
> >”My main concerns are about experiments. The authors mentioned in the introduction that several approaches can be used to incorporate the PDE parameters into the model, such as adding the PDE parameters as additional inputs, and include the parameters in the embedding module. However, it is not clear why these methods are not good. And the authors should compare with these methods in the experiments.”
>
> Thank you very much for your suggestion! We have addressed this point in our reply to all the reviewers.
>
> >”The proposed method contains more parameters than existing methods because of the introduction of the channel attention module. Therefore, the current comparison is not fair. The authors should train all models with the same number of FLOPs (or equivalently same training time).”
>
> Thank you for the feedback, due to space limitations the complexity of the model was reported in Appendix. To make this comparison clearer we created a new section B.1.1 with the previously reported table.
> As stated in the reply to all the reviewers and section 3 of the original submission, we have to emphasize that we set the model parameter size the same for both the base models with and without CAPE module. In fact, models with the CAPE module have fewer parameters in most cases. The concrete parameter number has also been provided in Table 6 in the appendix.
>
> > “I also have concerns about the proposed curriculum learning. The design of the proposed curriculum learning seems rather arbitrary. What is the intuition here? Especially for Eq. 12, I cannot see the benefit of this particular deign. The authors should elaborate more on this part. “
>
> >”From the results in Table 3, it doesn’t seem like the curriculum learning method is effective. For the advection equation, w/o curriculum learning has the best performance. And for the Burgers equation, the curriculum learning method also does not increase model performance.”
>
> Thank you for your kind comment. We added more experiments whose results are summarized in Table 8 of the Appendix. It showed the importance of our curriculum strategy in comparison to autoregressive and teacher-forcing methods.
> We also would like to emphasize that the basic motivation of our design of the curriculum strategy has been provided in the paragraph below equation 12. In addition, visualization and the algorithm of our curriculum strategy have been provided in appendix E of the original submission. Although we agree that a more sophisticated design of k_trans and the curriculum learning itself would be possible, we consider the present strategy to be simple and beneficial as it works well for the more challenging and realistic NS equations. We consider that finding a more sophisticated method is an interesting future work. Thank you very much again for your encouragement and helpful feedback.

---

> > ### Author Response · Authors · 2022-12-13
> > **Looking forward to your kind reply soon**
> >
> > Dear reviewer,
> >
> > We consider that every aspect about the paper that you criticized and asked about in your review was either already included in the original submission or was added with the updated version. We believe, therefore, that your score (reject) is in no way justified.
> > We look forward to kindly updating your score as soon as possible.

---

> ### Comment · Area_Chair_4JvV · 2022-12-13
> **Notice**
>
> Dear Reviewer,
>
> If you do not want to update your score, please at least acknowledge that you have already read the rebuttal.

---

### Official Review · Reviewer_65qq · 2022-11-04

**Confidence:** 3
**Clarity, Quality, Novelty And Reproducibility:** The paper is easy to understand and t…
**Correctness:** 4
**Technical Novelty And Significance:** 2
**Empirical Novelty And Significance:** 2
**Recommendation:** 6

**Strength And Weaknesses:**

Strength: The problem considered is interesting and the proposed method seems novel.

Weaknesses:
The biggest limitation from the reviewer's point is that the proposed CAPE module is not well motivated. More discussion is needed on why this module is introduced. The author mentions that the straightforward  method of including the PDE parameters as additional input will have negative impact on the accuracy of the network. However, no reference  is provided and  no experiments include this method as a baseline. The reviewer believes that the author should provide more references on the similar topic and add  more baseline methods for comparison, for example, $u^{k+1} = Base([u^k,\lambda])$ and $u^{k+1} = Base([u^{k-1},u^k,\lambda]).$

Also, more explanations are needed on the ablation experiments. It is clear that on the Advection and Burgers model, the method without curriculum learning is at least as good as the full method, and what is the advantage of using curriculum learning?

Update on Dec. 5: I do not have further questions.

**Summary Of The Paper:**

 This paper proposes a CAPE module which allows any data-driven SciML models to incorporate PDE parameters. The key idea is to  transform the input variables $u_k,\lambda$  into an  intermediate field data, and make the final prediction based on the original input and the intermediate output.

**Summary Of The Review:**

Please see Strength And Weaknesses.

---

> ### Author Response · Authors · 2022-11-14
> **Reply to Reviewer 65qq**
>
> Dear Reviewer 65qq,
>
> Thank you for your kind comment and your help in improving our paper. The following are our responses.
> >"The biggest limitation from the reviewer's point is that the proposed CAPE module is not well motivated. More discussion is needed on why this module is introduced.”
>
> Thank you for kindly pointing this out. We have modified a part of the introduction:
>
> “In this paper, we propose a new and effective parameter embedding module by utilizing the channel-attention method inspired by numerical solver with implicit method and style transfer in ML. (see Sec. 2.3 and Sec. 2.4)”
>
> We also emphasize that the motivation of the CAPE module structure has been provided in Sec 2.3 and Sec 2.4, in particular, in “CAPE as implicit discretization method” (paragraph) and “CAPE as style interpolation” (paragraph). Concerning the possible structure of CAPE module, we also added an ablation study of CAPE module’s structure in Section F and Table 9 of the appendix that show the present CAPE is not always the best but a better way overall.
>
> >"The author mentions that the straightforward method of including the PDE parameters as additional input will have negative impact on the accuracy of the network. However, no reference is provided and no experiments include this method as a baseline. The reviewer believes that the author should provide more references on the similar topic and add more baseline methods for comparison”
>
> Thank you for your kind suggestion. We have addressed this point of criticism in the general response to all reviewers.
>
> >”Also, more explanations are needed on the ablation experiments. It is clear that on the Advection and Burgers model, the method without curriculum learning is at least as good as the full method, and what is the advantage of using curriculum learning?”
>
> Thank you for pointing this out. We added more experiments including Unet in Table 8, which shows that our curriculum strategy is the best in most cases, though teacher-forcing worked quite well for the 1D Advection cases. Interestingly, only the curriculum strategy worked for 2D NS equations. Please note that the multi-dimensional NS equations are the actual physics equations with real-world applications, which is the ultimate goal of ML for science.
> For these reasons, we have concentrated on the 2D NS results in the main body and moved all the results to Table 8 of the Appendix.

---

> > ### Comment · Reviewer_65qq · 2022-12-13
> > **Reply**
> >
> > Thanks for replying. I have no further questions and have updated the scores.

---

> > > ### Author Response · Authors · 2022-12-13
> > > **Reply**
> > >
> > > Thank you very much again for your kindly checking our response!
> > > We really appreciate your fruitful comments and encourage to enhance our paper!

---

> ### Comment · Area_Chair_4JvV · 2022-12-13
> **Notice**
>
> Dear Reviewer,
>
> If you do not want to update your score, please at least acknowledge that you have already read the rebuttal.

---

### Official Review · Reviewer_WSSM · 2022-11-07

**Confidence:** 4
**Correctness:** 4
**Technical Novelty And Significance:** 2
**Empirical Novelty And Significance:** 2
**Recommendation:** 6

**Clarity, Quality, Novelty And Reproducibility:**

"An alternative approach attaches an external parameter embedding module to the network. However, there are too many
possible module structures and methods to provide the embedded parameter information to the base
network, and it is in general non-trivial to select the best one. Contrary to these ideas, we propose
a new and effective parameter embedding module by utilizing the channel-attention method" - It is unclear to me what is being said here.

**Strength And Weaknesses:**

Strength: A good number of PDEs are chosen to demonstrate the strengths (and weaknesses) of the CAPE method. A need for more parameter general neural PDE solvers is motivated and made clear.

Weakness: While other methods for incorporating PDE parameters are said to be insufficient, it would be useful to be comparisons against such naive methods. Additionally, ablations on the CAPE method would be useful to see which components of the architecture/curriculum learning strategy are the most critical. Also I would be curious to see a CAPE like method being used to augment pre-trained solvers, instead of the joint training required here.

**Summary Of The Paper:**

CAPE proposes a channel-attention-based method of generalizing existing neural PDE architectures to support training on equations with varying parameters. This is then combined with a curriculum learning strategy during training time. Evaluations are done on advection, Burgers, and 2d NS equations with Fourier Neural Operator and UNET being used as a baseline.

**Summary Of The Review:**

Generalization of PDE parameters is motivated as an important problem and the CAPE method using channel attention + curriculum learning is proposed as a solution which can be combined with existing solvers. However additional experimentation demonstrating the insufficiency of existing methods for generalization + some ablations is absent.

---

> ### Author Response · Authors · 2022-11-14
> **Reply to Reviewer WSSM**
>
> Dear Reviewer WSSM,
>
> Thank you very much for your kind comment for improving our paper. The following are our responses.
>
> >”While other methods for incorporating PDE parameters are said to be insufficient, it would be useful to be comparisons against such naive methods. Additionally, ablations on the CAPE method would be useful to see which components of the architecture/curriculum learning strategy are the most critical.”
>
> Thank you for your kind suggestions. We have addressed the above comments in our reply to “for all the reviewers/Comparison to conditional models”, and have added new experimental results to the updated manuscript (section F and table 9 in appendix).
>
> >”Also I would be curious to see a CAPE like method being used to augment pre-trained solvers, instead of the joint training required here.”
>
> Thank you for your very interesting suggestion!  This is a really interesting idea, highlighting one of the possible ways to take advantage of the inductive bias of the CAPE module, and we consider it as promising future work.
>
> >"An alternative approach attaches an external parameter embedding module to the network. However, there are too many possible module structures and methods to provide the embedded parameter information to the base network, and it is in general non-trivial to select the best one. Contrary to these ideas, we propose a new and effective parameter embedding module by utilizing the channel-attention method" - It is unclear to me what is being said here.”
>
> Thank you for kindly pointing out our insufficient explanation. We updated that part in the Introduction:
>
> “In this paper, we propose a new and effective parameter embedding module by utilizing a channel-attention method which is inspired by the implicit method of numerical solvers and style transfer approaches in ML (see Sec. 2.3 and Sec. 2.4)”.
>
> The point is that we would like to emphasize here that our CAPE module has a clear philosophy on its structure and implementation (“style transfer” and “implicit method”), different from typical conditional modeling which has infinite searching space of structure.

---

> > ### Comment · Reviewer_WSSM · 2022-12-13
> > **Update**
> >
> > Thank you for addressing my concerns. In light of a new comparison against a parameter-baseline I have updated my score.

---

> > > ### Author Response · Authors · 2022-12-13
> > > **Reply**
> > >
> > > Thank you very much for your kindly checking our reply!
> > > We really appreciate your very fruitful comments.

---

> ### Comment · Area_Chair_4JvV · 2022-12-13
> **Notice**
>
> Dear Reviewer,
>
> If you do not want to update your score, please at least acknowledge that you have already read the rebuttal.

---

### Author Response · Authors · 2022-11-14
**General Reply to All the reviewers**

Dear all the reviewers,

Thank you very much for your kind and encouraging suggestions! We have modified our paper following your advice, which has also been updated in OpenReview. The modified parts are colored in red. In this general response, we summarize our answers to common concerns shared by the majority of the reviewers.

## Importance of taking into account PDE parameters

First of all, we would like to re-emphasize the importance of taking into account PDE parameters. One of the main purposes of scientific machine learning is to solve PDEs. PDEs generally include a few PDE parameters controlling the strength of physical properties, such as diffusion, which are essential to describe the system's evolution and should be able to be treated in the ML models. Although a number of models have been proposed to tackle this problem, for example, FNO and Unet, almost all the models do not accept PDE parameters or consider only a very model-dependent method. To solve this problem, our paper proposed a universal method to take into account PDE parameters for the first time as far as we understood. We also would like to point out that our work provides a quantitive study on the generalization power of scientific ML models.
Note that the above information has been provided in our "Introduction".

## Comparison to Conditional Models
First of all, we found that all the reviewers asked for a comparison between our CAPE module and a  model that incorporates the parameters of the PDE. Since there is no unique way to define such a model, we call this class of model conditional, i.e. conditional to the PDE’s parameters.
Following your advice and, in particular, the method proposed by Reviewers 65qq and PGvV, we added new results for a conditional model that encodes the parameters as follows:The PDE’s parameters are concatenated to the input as an additional channel.
In Table 2 (also provided below), we have added the new results. The proposed CAPE method still has the lowest error. In particular, the CAPE module improves over the conditional baseline model by 30% on average.
$$
\\begin{array}{lllllll}
  \\hline
   PDE & model & Base &  Base (PINO) & Conditional & prev. 2-steps & CAPE  \\\
\\\
 \\hline
   \\\
   Advection & FNO &  0.69^{\\pm 2.2 \\times 10^{-3}} & 0.70^{ \\pm 1.6 \\times 10^{-4}} & 0.05^{\\pm 1.2 \\times 10^{-3}} & 0.13^{\\pm 3.9 \\times 10^{-2}} & {\\bf 0.04}^{\\pm 3.2 \\times 10^{-4}} \\\
\\\ \\\
         & Unet & 0.72^{\pm 1.0 \times 10^{-2}} & 0.76^{\pm 1.8 \times 10^{-2}} & 0.33^{\pm 2.0 \times 10^{-2}} & 0.33^{\pm 3.9 \times 10^{-2}} & {\bf 0.11}^{\pm 8.3 \times 10^{-3}} \\\
\\\ \\\
    & MPNN & 0.32^{\pm 2.5 \times 10^{-2}} & -- & 0.07^{\pm 2.0 \times 10^{-3}} & -- & -- \\\
\\hline
Burgers & FNO & 0.54^{\pm 0.40} & 0.49^{\pm 1.3 \times 10^{-1}} & 0.23^{\pm 2.3 \times 10^{-2}} & 0.17^{\pm 2.1 \times 10^{-2}} & {\bf 0.13}^{\pm 4.4 \times 10^{-3}} \\\
\\\ \\\
         & Unet & 0.53^{\pm 4.9 \times 10^{-2}} & 0.85^{\pm 3.2 \times 10^{-1}} & 0.51^{\pm 5.1 \times 10^{-2}} & 0.50^{\pm 6.3 \times 10^{-2}} & {\bf 0.45}^{\pm 3.8 \times 10^{-2}} \\\
 \\\ \\\
   & MPNN & 0.27 & -- & 0.13 & -- & -- \\\
\\\
\\hline
   2D NS & FNO & 1.06^{\pm 2.5 \times 10^{-2}} & -- & 0.86^{\pm 0.18} & 0.91^{\pm 0.10} & {\bf 0.80}^{\pm 8.3 \times 10^{-2}} \\\
 \\\ \\\
        & Unet & 0.77^{\pm 3.6 \times 10^{-2}} & -- & 0.74^{\pm 0.02} & 0.74^{\pm 3.5 \times 10^{-2}} & {\bf 0.70}^{\pm 3.7 \times 10^{-2}} \\\
\\\
\\hline
\\end{array}
$$
## Model Size
We would like to emphasize that the model size (number of parameters) is adjusted such that the models have nearly the same number of parameters. We provided this information in the original submission both in the main text (section 3, just above “Varying the parameter values”) and through table 6 in the appendix. The CAPE approach, therefore, performs better despite having approximately the same number of neural network parameters.

## Structure of CAPE module
As several reviewers pointed out, there are alternative module structures. We would like to emphasize that CAPE’s design choices for incorporating the PDE’s parameters are not a result of random search, but are well-motivated. We describe this in the submission on page 4 (top) in section 2.3 under the subsections “CAPE as style interpolation” and “CAPE as implicit discretization method” and further in section 2.4:


“The three convolutions can be interpreted as a finite difference method since convolution operations accumulate local information of a mesh, which, in principle, can simulate local interactions such as advection and diffusion. Intuitively, channel attention is equivalent to choosing an appropriate physical process for each PDE parameter.”

In addition, we added new results of the ablation study of the structure in Section F and Table 9 of the appendix. Overall, we believe that the concrete inductive bias of CAPE is well-motivated, and different design choices are thoroughly ablated.

---

> ### Author Response · Authors · 2022-12-06
> **Modification on Message-Passing PDE Solver results**
>
> Dear Reviewers,
>
> We have updated MPNN's results for full 3 trials. Note that CAPE is still better or comparable to MPNN which used a highly tuned and model specific parameter-embedding method.
> (standard deviation value is omitted in the case of Burgers equation because of length limitation)

---

### Comment · Area_Chair_4JvV · 2022-11-22
**Please respond as soon as possible if you still have questions on the paper.**

Please respond as soon as possible if you still have questions on the paper.

---

> ### Comment · Area_Chair_4JvV · 2022-11-29
> **Please respond to the authors by Nov. 30**
>
> Please indicate whether the authors' rebuttal addresses your concerns.
>
> If you still have questions, please ask as soon as possible.

---

> > ### Comment · Area_Chair_4JvV · 2022-12-05
> > **Zoom Meeting**
> >
> > For all reviewers, which have not responded to the authors, I will have to ask you to meet via Zoom. If you want to avoid such an additional step, please respond by Dec. 5.

---

### Author Response · Authors · 2022-11-29
**Please respond to our rebuttal**

Dear reviewers and AC,

It has been more than 2 weeks after our rebuttal and there have not been any reactions from the reviewers. As far as we can see, we have addressed all questions and criticism. In some cases, the information/experiments requested by the reviewers had already been part of the original submission. We would really appreciate some sort of acknowledgment of our work and comments.

Thank you!

---

### Public Comment · ~Shuhao_Cao1 · 2023-11-07
**Missing references**

Channel attention (that can be viewed as a learnable Galerkin projection) can be found in Cao NeurIPS 2021, and many follow-ups in this regard.

---

### Decision · Program_Chairs · 2023-01-20

**Decision:**

Reject

**Justification For Why Not Higher Score:**

NA

**Justification For Why Not Lower Score:**

NA

**Metareview: Summary, Strengths And Weaknesses:**

This paper introduces CAPE, a channel attention module that incorporates PDE parameters and can be integrated with any existing PDE solver. This module allows models to adapt to new PDE parameters and enhances inference efficiency. The authors also suggest using a curriculum learning approach to further improve model performance. The paper includes experimental results that demonstrate the effectiveness of the proposed method.

The reviewers raised several concerns on the experimental results: (1) The experiments only considered the training and testing sets and all hyper-parameters are directly tuned on the test set. There is a potential overfitting to the test sets. The authors need to use a validation set to tune the hyper-parameters, and the test set is only for calculating MSEs.  Therefore, the quality of the improvement cannot be assured, especially when the std is large (Table 2). (2) The curriculum leaning seems not effective either, especially considering it directly tuned over the test sets. From Table 8, the strategy is only effective for 3/6 of the experiments. (3) As can be seen from Appendix D.1, the proposed method is not very robust to the hyper-parameters. The results could be more convincing if the authors could improve the experiments in the next version.

---

> ### Author Response · Authors · 2023-01-20
> **Correction**
>
> Just in case someone reads the meta-review in the future, we want to mention that the points (1), (2), (3) in the meta-review above are demonstrably false. For instance regarding point (3), appendix D1 is about the parameters of the PDEs and *not* the hyperparameters of the proposed ML model.
>
> We have responded to each of these points before the meta-review was written but this response was ignored.
>
> If you are a reviewer of the future or a reader of this paper, please don’t take the points (1), (2), (3) to be correct.